



# Mapping photovoltaic power plants in China using Landsat, Random Forest, and Google Earth Engine

Xunhe Zhang[1,2,3], Shujian Wang[1], Yongkai Huang[1], Zunyi Xie[1,2], Ming Xu[1,2,3*]

[1]College of Geography and Environmental Science, Henan University, Kaifeng 475004, China

[2]Key Laboratory of Geospatial Technology for the Middle and Lower Yellow River Regions (Henan University), Ministry of Education, Kaifeng 475004, China

[3]Henan Key Laboratory of Earth System Observation and Modeling, Henan University, Kaifeng 475004, China

*Correspondence to*: Ming Xu (mingxu@henu.edu.cn)

**Abstract.** Photovoltaic (PV) technology, as an efficient solution for mitigating impacts of climate change, has been increasingly used across the world to replace fossil-fuel power to minimize greenhouse gas emissions. With the world's highest cumulative and fastest built PV capacity, China needs to assess the environmental and social impacts of these established photovoltaic (PV) power plants. However, a comprehensive map regarding the locations and extent of the PV power plants remains to be scarce at the country scale. This study developed a workflow combining machine learning and visual

interpretation methods with big satellite data to map the PV power plants in China. We applied a pixel-based Random Forest (RF) model to classify the PV power plants from composite images in 2020 with 30-meter spatial resolution on Google Earth Engine (GEE). The result classification map was further improved by a visual interpretation approach. Eventually, we established a map of PV power plants in China by 2020, covering a total area of 2917 km$^2$. Based on the derived national PV map, we found that most PV power plants were sited on cropland, followed by barren land and grassland. In addition, the

installation of PV power plants has generally decreased the vegetation cover. This new dataset is expected to be conducive to policy management, environmental assessment, and further classification of PV power plants.

## 1 Introduction

Solar power is the most available renewable energy source with a great potential to replace fossil fuels for reducing greenhouse gases (GHGs) emissions and mitigating climate change (Nemet, 2009; Creutzig et al., 2017). Photovoltaic (PV)

technology can convert solar energy directly into electricity with large arrays of solar panels. With the fast development of PV technology and industry, the cost of electricity generated by PV power plants has declined to the same level as the traditional fossil-fuel power plants (Zou et al., 2017). However, the development of PV power plants takes up a large amount of land (Capellán-Pérez et al., 2017). Potential environmental impacts can be induced during the process of construction and operation





of PV power plants, such as changes in local microclimate (Taha, 2013; Barron-Gafford et al., 2016; Yang et al., 2017; Chang
et al., 2018; Broadbent et al., 2019), albedo (Nemet, 2009; Li et al., 2017; Zhang and Xu, 2020), vegetation cover (Liu et al.,
2019; Nghiem et al., 2019),  land cover  (Fthenakis and Kim, 2009; Hernandez et al., 2014a; Hernandez et al., 2015), and
habitat biodiversity etc (Turney and Fthenakis, 2011; Hernandez et al., 2014b). Therefore, it is urgently needed to evaluate
these effects from the rapidly growing PV power plants and make planning recommendations. China's PV industry leads the
world in cumulative installed capacity and new installed capacity. According to the National Energy Administration of China,
by the end of 2020, China's total installed solar capacity had reached 252.8 Gigawatt (GW), with 48.2 GW being newly
installed in 2020, most of which was from photovoltaic (PV) power. In 2020, China declared to aim to achieve emissions peak
before 2030 and achieve carbon neutrality before 2060. This aim will lead to greatly increase in PV power plants across China.
However, data regarding the distributions of PV power plants remained to be scarce in China at the country scale.
Consequently, the ability of rapidly, precisely, and virtually mapping PV power plants is essential for national policy
management and environmental assessment.

Machine learning has been widely used in the remote sensing community as a practical empirical approach for regression
and classification (Lary et al., 2016). Remote sensing techniques can acquire features of different ground objects in spectral,
temporal, and spatial dimensions (Zhu et al., 2012). Machine learning algorithms can generally model complex class signatures
with high accuracy by incorporating various input predictor data such as spectral, texture, spatial information without data
distribution assumptions (Maxwell et al., 2018). Machine learning methods have been applied to map PV power plants with
various remote sensing images on regional scales (Malof et al., 2016a; Malof et al., 2016b; Malof et al., 2017; Hou et al., 2019;
Jie et al., 2020). However, as PV power plants have been increasingly built in various landscapes worldwide, such as deserts,
mountains, coast, lakes (Sahu et al., 2016; Al Garni and Awasthi, 2017; Hammoud et al., 2019), which makes accurately
identifying the PV power plants on a continental scale challenging due to their mixture of PV panels, shadows and variety land
types.

Training machine learning models require numerous training samples, and visual interpretation is still widely used to cover
as much system parameter space as possible. Bradbury et al. (2016) created a solar PV arrays dataset of four cities in California,
USA, by manually annotating the PV arrays boundaries. Dunnett et al. (2020) developed a global solar plants dataset by
collecting the features annotated as solar power from OpenStreetMap, a free editable and open-access map (Haklay and Weber,
2008). Deep learning methods, including convolutional neural networks (CNNs), recurrent neural networks (RNNs),  have
been applied to map the PV power plants in the United States (Yu et al., 2018), China (Hou et al., 2019), and worldwide
(Kruitwagen et al., 2021) with sufficient hand-label data. Deep learning models can provide high accuracy but need extensive
computation resources. Furthermore, the labelled data used to train the deep learning model cannot ensure that it will cover
the majority of the system parameter space for identifying PV power plants in a complicated geographical and atmospheric



environment. On a continental scale, the small total area of PV power plants and large background area of non-PV power plants will lead to a relatively high commission error. Visual interpretation is a method to guarantee the high quality of classification results and provide extra training data for the following machine learning tasks.

In this study, we integrate the advantage of machine learning and visual interpretation to map the PV power plants in China. We used Google Earth Engine (GEE), a cloud geospatial computing platform that supports petabyte remote sensing data and
multiple machine learning algorithms (Gorelick et al., 2017), to acquire the preliminary classified result by machine learning model with Landsat imagery. With GEE's support, researchers in the remote sensing community have completed numerous classification works on a continental planetary scale (Gong et al., 2020; Xie et al., 2019; Li et al., 2019; Gong et al., 2019; Deines et al., 2019; Goldblatt et al., 2018; Pekel et al., 2016; Dong et al., 2016; Lobell et al., 2015; Hansen et al., 2013; Mao et al., 2021). We further assessed the climate and geographic information for the PV power plants in China. This information
is critical for policymaking, environmental assessment, and site selection for PV power facilities.

In summary, the objectives of this study are to (1) build a workflow to map the PV power plants on a continental scale with Landsat imagery on GEE; (2) produce a fine-resolution map of PV power plants in China and (3) analyze the distribution characteristics of PV power plants in China.

## 2 Materials and Methods

## 2.1 Machine Learning Classification

### 2.1.1 Landsat−8 surface reflectance imagery

This study used the Landsat-8 (L-8) surface reflectance (SR) product with a 30 m spatial resolution. L-8 product has been atmospherically and topographically corrected and is accessible on GEE. Using the pixel quality control bands, we removed the pixels contaminated by clouds and shadows in each image. We further composited L-8 image datasets using the median
value of six reflective bands during a specific period. The composite image was robust against extreme values and provided enough information about the particular period (Flood, 2013). We composited the images of autumn 2020 (September to November) and of the whole year of 2020 (January to December) over China, respectively. The composite image in autumn (C1) has the advantages of fewer clouds, snow, and vegetation in China compared to the image from other seasons. The composite image of the whole year (C2) was involved in nearly four times as many images as the C1, so the C2 is less affected
by the contaminated pixels than C1 and has less timeliness. Therefore we used C2 as a substitute in the regions where the quality of C1 was poor.



Earth System
Science
Data

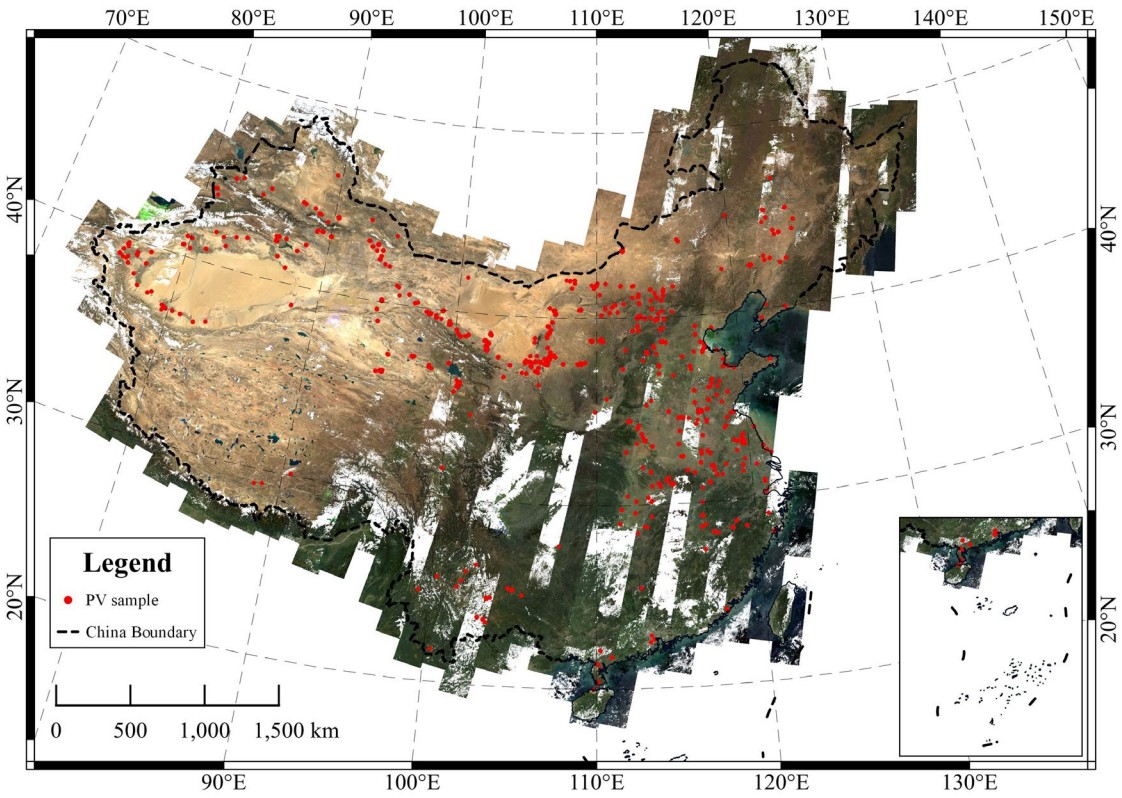

**Figure 1.** The composite image from Landsat-8 imagery during autumn 2020 and PV samples of training and validation set in this study.

### 2.1.2 Random forest classification

We used a pixel-based Random forest (RF) algorithm on GEE to map the PV power plants over entire China. The RF classifier is an ensemble classifier that uses a set of decision trees to predict classification or regression with advantages of high precision, efficiency, and stability (Belgiu and Drăguţ, 2016). The RF classifier has also been proven to be better than other machine learning classifiers on GEE (Zhou et al., 2020; Phalke et al., 2020) for mapping rangelands and croplands. For the RF classifier, we set the number of trees to 500 and left the rest of the parameters at GEE's default. Compared with the

object-based model classification, the pixel-based model classification uses the raw resolution pixel and does not require further segmentation of the classified image.

### 2.1.3 Training and validation samples

The RF classifier is sensitive to the sampling design (Belgiu and Drăguţ, 2016). Suitable training samples could ensure an RF trained model's classification accuracy and stable performance. We collected and labelled sample data as PV region and

non-PV region, short as PV and NPV, respectively. We primarily collected the PV samples from Dunnett's dataset, a global

solar plants dataset annotated by volunteers (Dunnett et al., 2020). We manually modified this dataset with Google Earth's background to ensure the PV samples locating inside the PV power plants. We also manually selected and edited the extent of different PV power plants which not annotated in Dunnett's dataset. We stored all the PV samples as polygon vectors. In total, the area of the PV sample polygons was 1121 km$^2$. We randomly sampled points within the polygons with a balanced quantity from humid and arid regions (Fig. 1).


We collected the NPV samples from adjacent regions of the PV power plant region within 5-kilometers buffer regions, the samples from manfully selected typical land types, and the samples from the whole of China, respectively. In total, we prepared 20000 points labelled as PV and 50000 points labelled NPV in this study. At last, after filtering out the low-quality pixels, we randomly chose 75% of the total points as the training set and the left 25% of the total points as the validation set (Table 1).

**Table 1.** Training and validation dataset

| Set | PV for Training | non-PV for training | PV for Validation | non-PV for validation |
|-----|-----------------|---------------------|-------------------|-----------------------|
| C1 | 15508 | 34780 | 4874 | 11850 |
| C2 | 15022 | 37605 | 4978 | 12257 |

Note: Composite image one (CS1) is composited from Landsat images during 2020.9-2020.11

Composite image two (CS2) is composited from Landsat images during 2020.1-2020.12

### 2.1.4 Calculation of variables

We collected nine variables from the Landsat-8 SR images data, including six original bands and three calculated indexes
(Zhang et al., 2021b). We used these variables to train machine learning models to distinguish the PV and non-PV regions. The six original bands included blue (B2), green (B3), red (B4), near-infrared (B5), two shortwave infrared bands (B6 and B7) from the Landsat-8 images. The three indices included the Normalized Difference Vegetation Index (NDVI) (Tucker, 1979), the Normalized Difference Built-up Index (NDBI) (Zha et al., 2003), and the Modified Normalized Difference Water Index (MNDWI) (Xu, 2006).

### 2.1.5 Classification accuracy assessment

We evaluated the pixel-based RF model by using a validation set. By comparing the confusion matrix of categorized and labelled points in the validation set, we used the kappa coefficient, overall accuracy, producer's accuracy, and user's accuracy of the validation set to assess the performance of the model (Congalton, 1991). The kappa coefficient calculated from the confusion matrix is widely used to check consistency and evaluate model performance. The overall accuracy is measured to

examine the overall efficacy of the model. The producer's accuracy indicates the proportion of truth samples correctly judged as the target class. The user's accuracy indicates the proportion of samples judged as the target class on the classification map present as truth samples.

## 2.2 Visual interpretation

### 2.2.1 Filter and morphological operations

By applying the RF classification, we got pixels categorized as PV region and NPV region over entire China. We filtered the pixels by topography. The PV power plants are not suitable for being built in locations with large slopes and shady slopes (Al Garni and Awasthi, 2017; Aydin et al., 2013). We calculated slope and hillshade from the Shuttle Radar Topography Mission (SRTM) with 30 m spatial resolution (Farr et al., 2007). We calculated the hillshade by setting azimuth as 180° and elevation angle as 45°. We filtered the pixels where slope larger than 30° and value of hillshade less than 150.

In pixel-based classification, sudden disturbances in the image signal and different objects with the same spectrum or the same objects with a different spectrum can cause a salt-and-pepper noise (i.e., impulse noise) which presents as image speckles. Additionally, the edge of the PV power plants mixed with road or other PV facilities that are not categorized as PV regions should be part of the PV power plants. We filtered categorized PV pixels that connect less than 9 pixels to neighbors to reduce the salt-and-pepper noise. We then used morphological operations on the GEE platform to dilate the PV pixel clusters. The

morphological operations included one round max filter and one round mode filter with a circle kernel of one-pixel radius to conduct spatial filtering.

### 2.2.2 Visual interpretation

We further convert the clusters of PV pixels into polygonal vectors on GEE. We used visual interpretation to identify all polygons categorized as the PV power plants by the RF model. To meet the visual interpretation needs, we calculated each

polygon's areas and filtered them with less than 0.04 km$^2$, which equaled 45 adjacent pixels. According to Kruitwagen's dataset, PV power plants with an area of more than 0.04 km$^2$ account for 94.2 percent of the total area of PV power plants in China (Kruitwagen et al., 2021).

With QGIS software (http://www. qgis. org/) and the GEE plugin (https://gee-community.github.io/qgis-earthengine-plugin/), we filter the PV polygons with visual interpretation based on their sizes, shapes, color, and texture with background

true-color images from Landsat-8, Sentinel-2, and Google Earth (Fig. 2). We first collected the PV power plants from the classified result of CS1, which stood for the image in autumn of 2020, and we then collected the PV power plants from the result of CS2, where clouds still contaminate CS1.

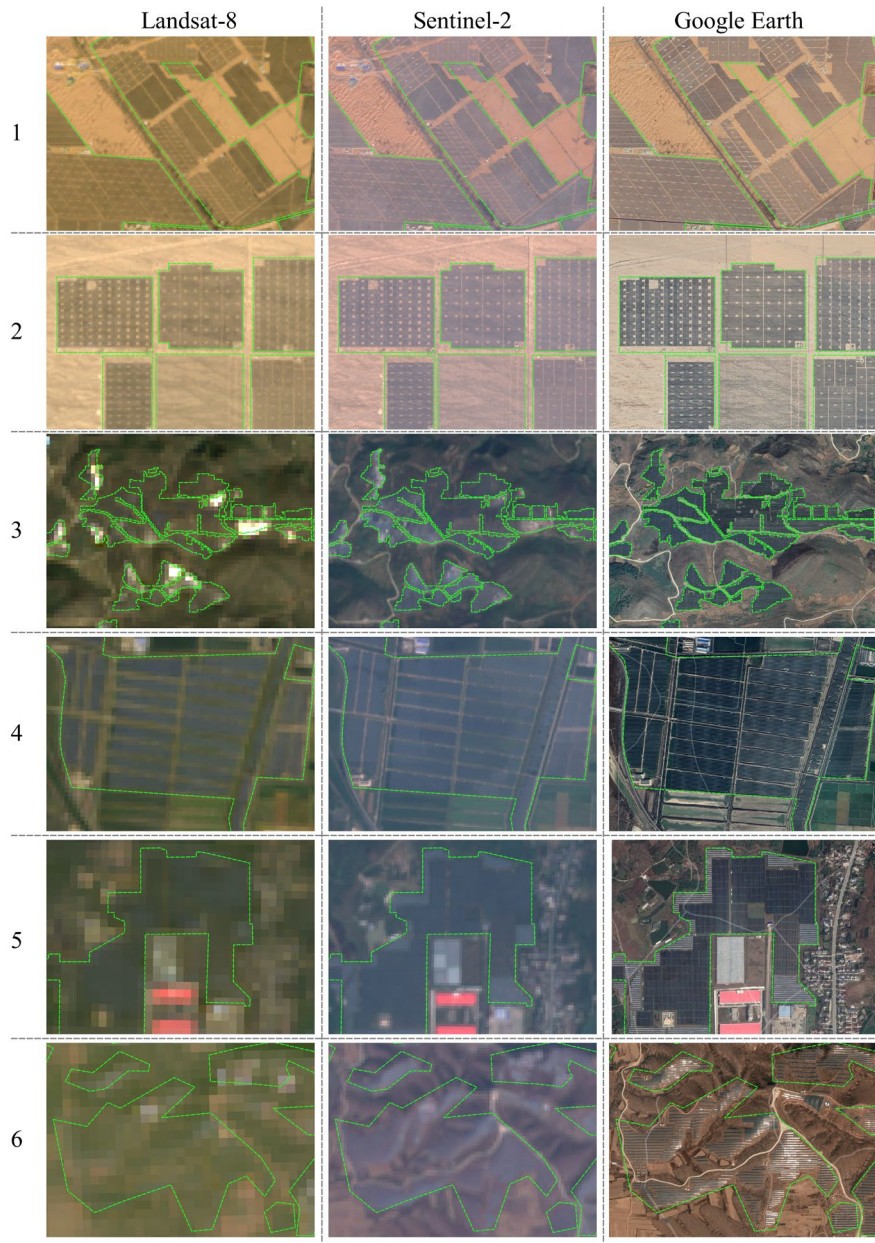

**Figure 2.** The visual interpretation examples from Landsat-8, Sentinel-2, and Google earth RGB true-color images. The green dashed line
is the boundary of PV panels. © Google Earth 2021.




## 2.3 Dataset organization and statistical analysis

We showed the flowchart of this study (Fig. 3). We also mapped some regions containing PV power plants as examples to show the changes of different steps (Fig. 4).

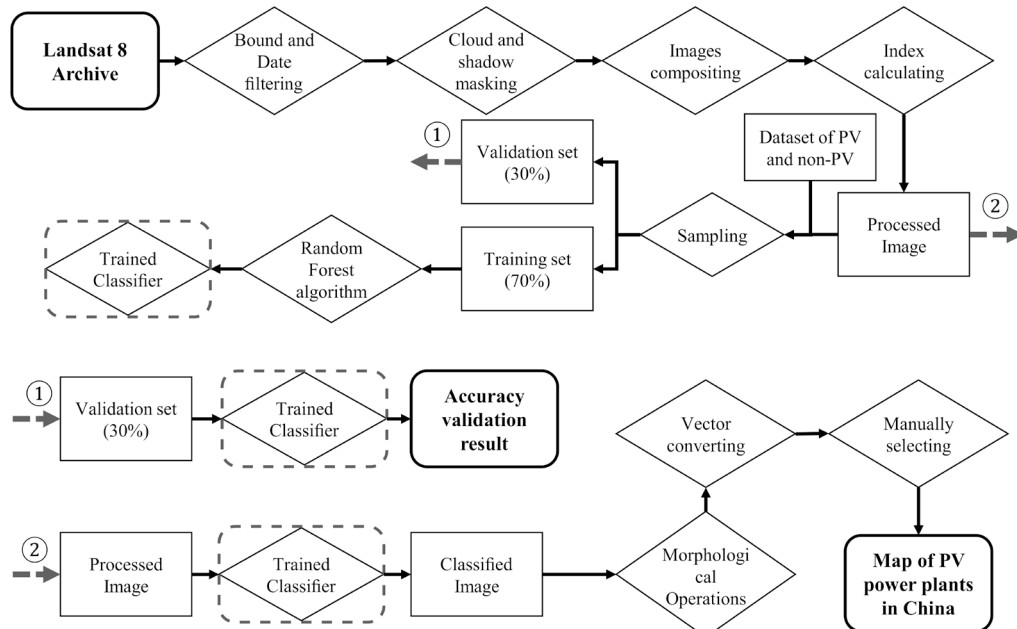

**Figure 3.** The flowchart of mapping the PV power plant in China.

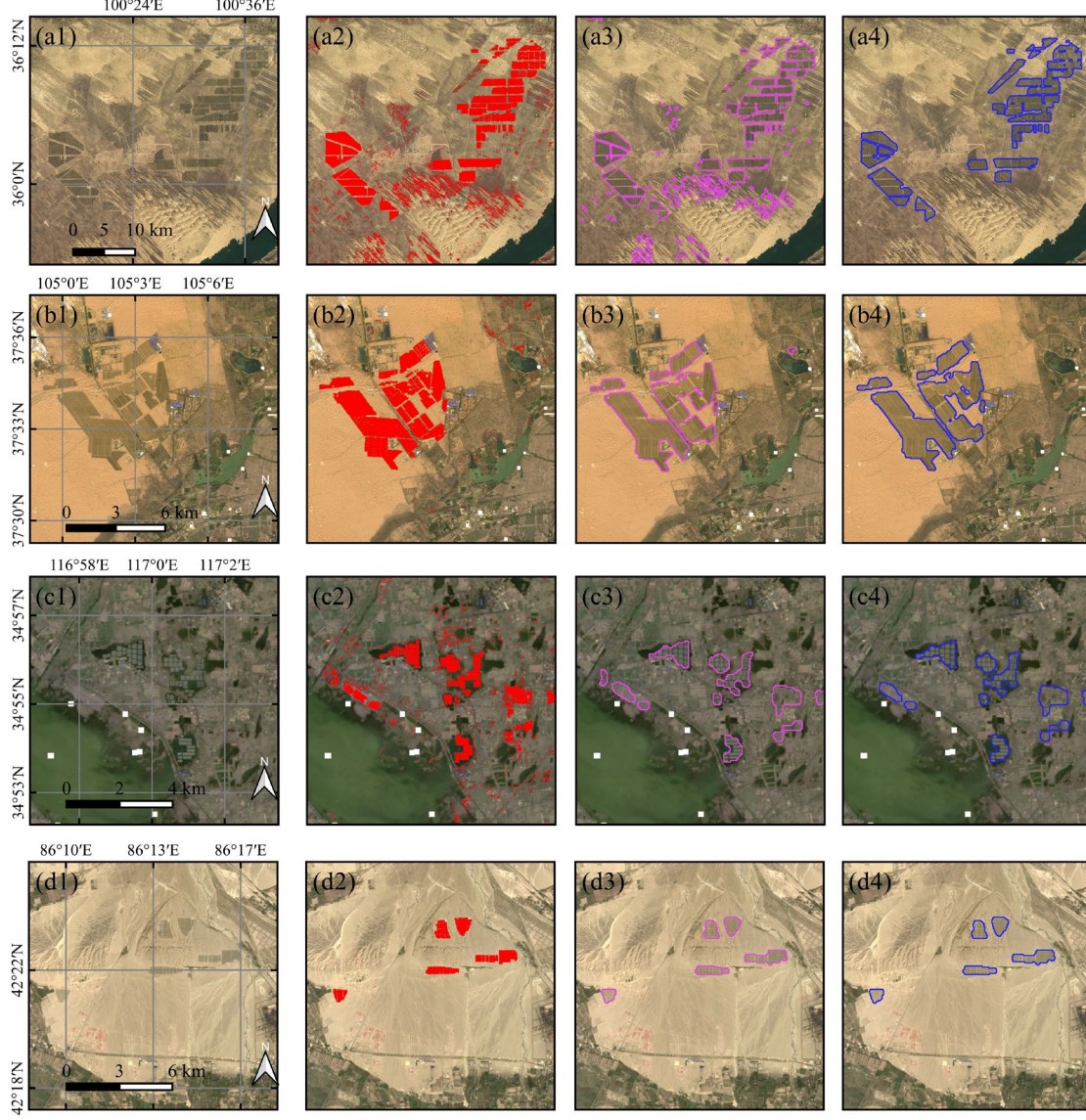

**Figure 4.** The examples of different steps (a-d 1) true-color of Landsat-8 composite image in autumn of 2020, (a-d 2) random forest classification result in red color (a-d 3), result in pink color after filtering, morphological operations and vector converting, (a-d 4) result in purple color after manually selecting and improving.

We built a dataset of PV power plants in China. We stored the PV power plants as polygon objects with shapefile format (Falge et al., 2017). Since PV power plants are not entirely adjacent, we group the PV power plants within 10 kilometers for further analysis. We further calculated the area, average elevation, annual mean air temperature, cumulative yearly



precipitation, population density, annual mean enhanced vegetation index (EVI), and land cover type of each PV power plant (Table 2). All the datasets are available on GEE.

**Table 2.** The attribute of the PV power plants.

| Attribute | label in dataset | Data source | | Data spatial resolution | Calculated method | Periods |
|---|---|---|---|---|---|---|
| Average elevation | elev | SRTM | Farr et al. (2007) | 30 meter | Mean value within an object | 2000 |
| Annual mean temperature | temp | ERA5 | Service (2017) | 0.25 Degree | Value from object centroid | 1990 to 2020 |
| Annual precipitation | precip | ERA5 | | 0.25 Degree | Value from object centroid | 1990 to 2020 |
| Population density | popu | WorldPop | Tatem (2017) | 100 meter | Mean value from object 100-kilometers buffer | 2020 |
| Annual mean EVI in 2013 | EVI 2013 | Landsat-8 EVI | Roy et al. (2014) Huete et al. (2002) | 30 meter | Mean value within an object | 2013 |
| Annual mean EVI in 2020 | EVI 2020 | Landsat-8 EVI | | 30 meter | Mean value within an object | 2020 |
| Land cover type | landcover | ESA WorldCover | Zanaga (2021) | 10 meter | Mode value from object 2-kilometers buffer | 2020 |

## 3 Result

The map indicating the distributions of the PV power plants in China is shown below (Fig. 5a). The total area of the PV power plant derived in this study was 2917 km$^2$ by the autumn of 2020. In the machine learning classification process, the result showed that the model with the dataset of CS1 had a comparable result with the model with the dataset of CS2 (Table

3). The kappa coefficient (kappa), overall accuracy (OA), user's accuracy (UA) of PV and non-PV (NPV), and producer's accuracy (PA) of PV and non-PV were 0.878, 95.04%, 95.51%, 93.82%, 97.59 and 88.83% for the CS1. The kappa, OA, UA of PV and NPV, and PA of PV and NPV were 0.886, 95.39%, 95.961%, 93.89%, 97.62, and 89.89% for the CS2, respectively (Table 3).



**Table 3.** Validation parameters for the model trained model with different variables sets.

| Image | Kappa | OA (%) | UA NPV (%) | UA PV (%) | PA NPV (%) | PA PV (%) |
|---|---|---|---|---|---|---|
| C1 | 0.878 | 95.04 | 95.51 | 93.82 | 97.59 | 88.83 |
| C2 | 0.886 | 95.39 | 95.96 | 93.89 | 97.62 | 89.89 |

180          Note: kappa coefficient (Kappa), overall accuracy (OA), producer's accuracy (PA), and user's accuracy (UA).

The summed area of PV power plants in all provinces showed that the top three provinces in installation PV power plants were Qinghai, Xinjiang, and Inner Mongolia, respectively (Fig. 5b). The result based on the land cover also showed that most PV power plants were sited on cropland, followed by barren land and grassland (Fig. 5c).





**Figure 5.** (a) The distribution and the heat map of the PV power plants in China, (b) the areas of PV power plants in each province of China, (c) the areas of PV power plants by the land cover of China.

We have further counted the distributions of PV power plants by temperature, precipitation, elevation, population density, and location. From the result, many PV power plants are located in China's arid and alpine region, where solar energy resources are plentiful, precipitation is low, vegetation is sparse, population density is low, and elevation is relativity high (Fig. 6). Additionally, some PV power plants are located in the industrially developed eastern coastal provinces of China, where precipitation is high, density population is high, and elevation is low. This distribution result also shows two tendencies in

China's site selection of PV power plants. One is to lean toward installing in areas with suitable natural conditions but less power demand. The other is to lean toward installing in the areas with more local energy demand.

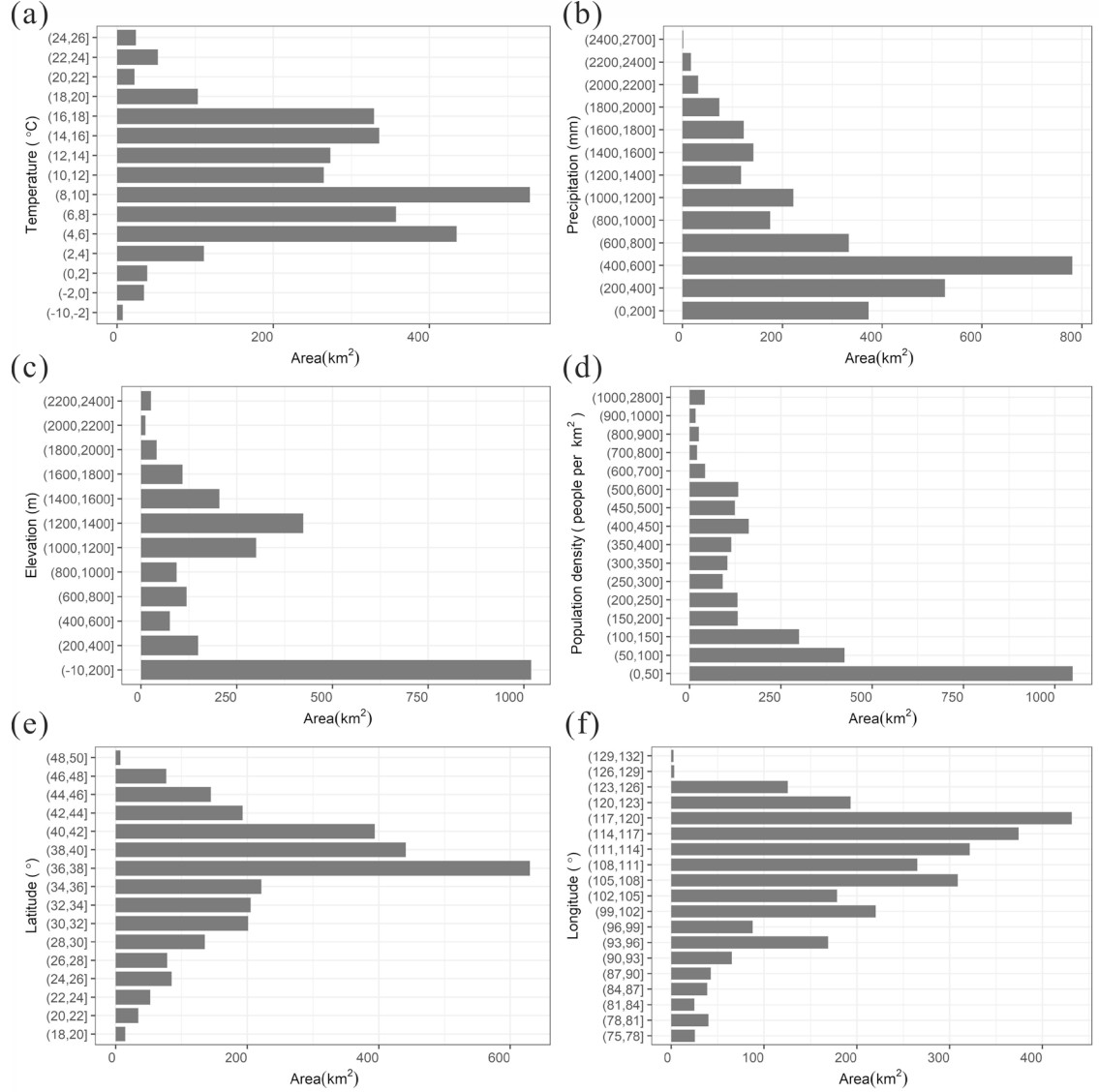

**Figure 6.** The area of PV power plants is counted by (a) temperature, (b) precipitation, (c) elevation, (d) population density, (e) latitude and longitude.

Installation of PV power plants affects the local vegetation under different climate conditions (Zhang and Xu, 2020; Nghiem et al., 2019; Liu et al., 2019). We calculated and compared each PV power plant's annual mean EVI (larger than 0) in 2013 and 2020 from Landsat-8 images. By the record of National Energy Administration of China, the cumulative installation



of PV capacity is 19.4 GW by 2013 and 252.8 GW by 2020, which indicates that over 92% of PV power plants are installed
       after 2013. Comparing the EVI between 2013 and 2020 over PV power plant areas, we discovered that the EVI values of PV
       power plants in 2020 was strongly and positively linked with the that in 2013, of which the linear regression with area weight
       ($p < 0.01$) showed the estimated slope was 0.594 and intercept was 0.0312 (Fig. 7). From the linear model result, we found
       that the installation of PV power plants generally decreased the EVI in regions of high vegetation cover. By contrast, in the
hyper-arid regions, where EVI was less than 0.07, the installation of PV power plants slightly increased the EVI values.

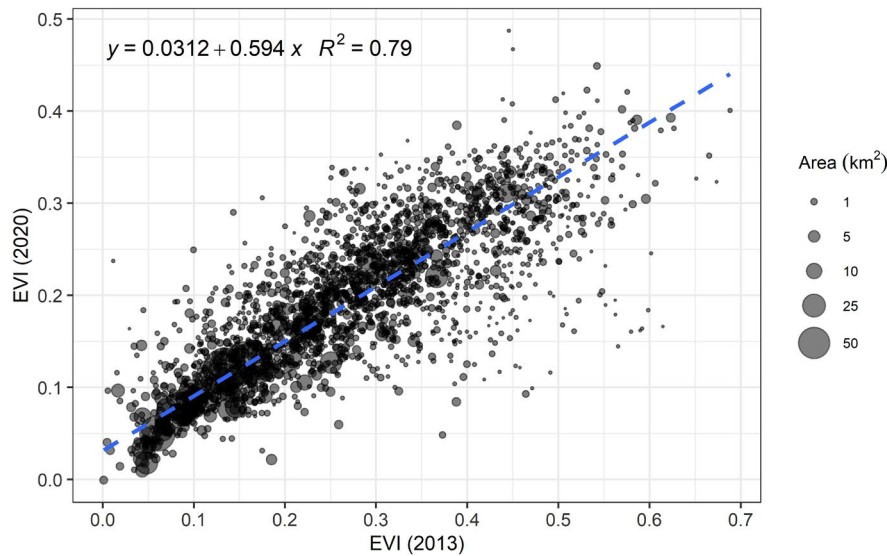

**Figure 7.** EVI values of PV power plants in 2020 vs those in 2013 across China.

## 4 Discussion and conclusion

       Photovoltaic technology is one of the essential technologies to obtain solar power globally. Information on the spatial
extent of PV power plants is critical for policy management and environmental assessment. We developed a dataset that
       collects the distribution for PV power plants in China with a spatial resolution of 30 m by the autumn of 2020. The PV power
       plant regions were categorized based on machine learning using Landsat-8 imagery on GEE. To our knowledge, this is the
       latest and most complete public dataset for the spatial extent of PV power plants in China. In the previous study, Dunnett et
       al. (2020) also provided a harmonized solar plants dataset obtained from an open-access map containing many PV power plants
in China. However, the comparison between the two datasets suggested that their dataset of PV power plants relies on voluntary
       annotation is incomplete and with no guarantee on update timely in China.



Our method integrates the efficiency of machine learning and the accuracy of visual interpretation. The two pixel-wise RF models performed well, with PV producers' accuracy over 84%. PV power plants are a mixture of PV panels and their occupied lands, which often cause challenges in mapping PV power plants. As PV power plants are distributed under various climatic conditions and land cover types in China, the PV power plants are more likely to have the similar spectral features as other objects, such as plastic-cover shed and biological soil crust. PV power plants in different regions have different PV panel spacing and tilt angles due to the sunlight incident angle and terrain, which could cause spectral variability (Yadav and Chandel, 2013; Ji et al., 2021). The model trained by large and scattered training samples ensures that most PV power plants are successfully identified in China under various conditions. Nevertheless, there are still have some omission errors in the RF classification result. Misclassified PV regions with sporadic distribution among the PV power plants will not impact the morphological operations and visual interpretation results. However, some PV power plants, which are of the low density of PV panels, would be misclassified as non-PV objects as a whole. In particular, these PV power plants situated in mountainous areas typically have unique installation spacing and installation angles for their solar panels. Additionally, the mountainous terrain also impacts the reflectance of the PV power plants (Wen et al., 2018). These PV power plants thus mainly were missed in our study but only took up a small portion of the total number.

While the overall accuracy of the two RF models is about 96%, the misclassification of PV power plants with commission error was much higher than the entire total area of the PV power plants in China with a vast classified area. After transferring the pixel clusters to objects of vectors, we have spent dozens of hours of visual interpretation work filtering the misclassification regions with commission error. This visual interpretation process could improve the quality of the map classified by the RF model. Object-based methods with machine learning models can identify the target feature accuracy and efficiency (Blaschke, 2010; Desclée et al., 2006; Xiong et al., 2017). However, object-based methods with machine learning models still need plenty of training samples to improve models' accuracy. Our result with vector format could provide the training samples for researchers to identify PV power plants based on object-based methods in the future.

Based on the derived national PV map, we found that PV power plants are more likely to be installed in areas with suitable natural conditions but low power demand, or in areas with high local energy demand. This is due to the fact that lands are more productive and thus more valuable in areas with higher population density and higher demand for power. Additionally, the installation of PV power plants will generally decrease the vegetation. Thus, PV development needs are in conflict with the land use costs.

In this study, we have successfully established a dataset for PV power plants with a total area of 2917 km$^2$ in the whole of China until 2020. This dataset is conducive to policy management and environmental assessment. This dataset may also support potential PV power plants to research as training samples.



## 5 Data availability

The dataset of photovoltaic power plant distribution in China by 2020 is stored as shapefile format and available to the public at https://doi.org/10.5281/zenodo.4552919 (Zhang et al., 2021a).

**Author contribution**

X. Zhang and M. Xu designed the research, performed the analysis; X. Zhang wrote the paper; X. Zhang and J. Wang performed the analysis; Z. Xie edited and revised the manuscript; X. Zhang, J. Wang and Y. Huang prepared the data.

## Competing interests

The authors declare that they have no conflict of interest.

**Acknowledgements**

This research has been supported by the funding of the National Key Research and Development Program of China (2017YFA0604300, 2018YFA0606500).

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
