# Peer review of "Mapping photovoltaic power plants in China using Landsat, Random Forest, and Google Earth Engine"

_Earth System Science Data, 2022_

## Author Response (AR1)

Author Response to the Reviewer Comments to the manuscript "Mapping photovoltaic power plants in China using Landsat, Random Forest, and Google Earth Engine" [essd-2022-16] submitted to Earth System and Science Data.

We thank two anonymous referees for their constructive reviews, by following which the manuscript has substantially improved. The individual comments are listed below (shown in blue) including our responses (shown in black). The changes discussed in this reply will be included in the revised manuscript and will thus become visible after re-submission.

**RC1: 'Comment on essd-2022-16', Anonymous Referee #1 and our response**

In this manuscript, authors developed a workflow combining machine learning and visual interpretation methods to map the Photovoltaic power plants in China. This topic is very important to assess the environmental and social impacts of these estabilished photovoltaic power plants. In fact, there are a number of papers on remote sensing target information extraction, the paper is not particularly novel. And, there are some problems in this paper:

Response: We are grateful for referee #1's recognition of this study's importance. Although there are a number of previous studies for mapping land properties, PV power plant mapping has not been widely conducted, and there still lack the open dataset for PV power plant in China. Our dataset offers the latest public dataset for the spatial extent of PV power plants in China. In this study, we integrate the advantage of cloud computing, machine learning, visual interpretation and freely available remote sensing imagery to map the PV power plants in China.

1, The introduction is inconformity with the objectives of study. For example, the one and two paragraphs are talking about the Photovoltaic power plants, machine learning, which can be wrote in a more refined way and introduce the main topic quickly. The references in the Introduction section are too limited, the authors should refer more works of relation analysis on deep learning methods, PV power plants and Remote sensing images, about the mechanism of deep learning and remote sensing image extraction of PV power plants, this paper hasn't given more details description.

Response to comment 1: Following referee #1's suggestion, we have rewritten our introduction part to fit the objectives of this study. We have streamlined the introduce part. We added sentences to describe the mechanism of machine learning and deep learning. We also added more references about learning. We further explained the advantages of our methods.

2, Figure 2 (page 7):  the specific meanings of 1, 2, 3, 4, 5, and 6 in this figure should be explained. Response to comment 2: We are sorry for the unclear description of Figure 2. We have refined the figure 2 with accurate description. The specific meanings of 1, 2, 3, 4, 5, and 6 are the 6 example sites to show the true-color images from Landsat-8, Sentinel-2, and Google Earth for visual interpretation.

3, There are too many texts in the discussion section and need to be further streamlined.
Response to comment 3: We have streamlined and shortened the discussion in the revised manuscript.

4,In addition, did the authors consider how to validate the results? how can we believe the results? e.g. what validation, more specific about ground truthing etc. Without this information, I can not trust the results of this paper.
Response to comment 4: In this study, there were two stages for mapping the PV power plants in this study. In the first stage, we used a pixel-based random forest model with selected features to map the PV power plants in China. We further validated the model and the performance of the random forest model using kappa coefficient, overall accuracy, producer's accuracy and user's accuracy (Table 3). In the second stage, we used visual interpretation method to filter the misclassified PV power plant due to commission errors in machine learning process. We carefully inspected each polygon with the latest Landsat-8, Sentinel-2, and Google Earth true color images by visual interpretation. The entire visual interpretation step took us about 2 weeks. While visual interpretation is time consuming, it generally offer validation with high accuracy.

In the revised manuscript, we added extra validation by comparing our dataset with the Dunnett's dataset and Kruitwagen's dataset in China. Dunnett et al. (2020) provided a harmonized solar plants dataset obtained from OpenStreetMap, an open-access map. The PV power plants in the open-access map were annotated by volunteers. The total area of PV power plants in China from Dunnett's dataset is 897.4 $km^2$, of which 842 $km^2$ have spatially intersected with our dataset. The no intersected solar panels area is 55.4 $km^2$. Some of them are too small for our method to recognize.

Kruitwagen's dataset (Kruitwagen et al., 2021) was classified by deep learning methods. The total area of PV power plants in China from Kruitwagen's dataset is 2169.8 $km^2$ by 2018, of which 1873.5 $km^2$ have spatially intersected with our dataset. The PV power plants in Kruitwagen's dataset that do not intersect with our dataset are 296.3 $km^2$, some of which are too small to be identified by our method and some of which are misidentified in Kruitwagen's dataset.

We found our methods could fail sometimes to recognize these PV power plants situated in mountainous areas that typically have unique installation spacing and installation angles for their solar panels. Small size PV power plants (less than 0.04 $km^2$) was potentially another reason for mis-classification in this study.

**RC2: 'Comment on essd-2022-16', Anonymous Referee #1 and our response**

In this manuscript, machine learning and visual interpretation methods were combined to map the PV power plants in China. The topic is very important, and the study results would be useful for developing PV industries in the future. However, there are some problems in the manuscript which should be solved before reconsideration for publication in Earth System Science Data:
Response: We thank referee #2 very much for the positive feedback.

1. The Introduction section should be written. The novelty of this study compared to the previous studies should be highlighted. And the useless contents should be removed.
Response to comment 1: By following referee #2's comment, we have rewritten the Introduction. In the new Introduction, we introduced the advantage of cloud computing, machine learning, visual

interpretation and freely available remote sensing imagery to map the PV power plants in China. We also highlight the advantages of our study compared with previous studies in the introduction part. The current Introduction has been largely refined.

2. The "Dunnett's dataset" was used as the basic training and validation samples, which means your model's ability was "limited" to this dataset. In other words, the PV power plants that cannot be identified in "Dunnett's dataset" may also be ignored in your model. Although you mentioned that you modified the dataset, how it was implemented and what is the difference between the modified dataset and the original dataset were not clearly stated.

Response to comment 2: We thank referee #2 for pointing out this. Suitable training samples are indeed crucial for an RF model's classification accuracy and stable performance. We modified the Dunnett's dataset as our training samples in this study, and found that the labelled PV power plants in Dunnett's dataset are rarely distributed in eastern China, which will limit our model's performance to identify the PV power plant in similar areas. So we further manually selected and edited the extent of different PV power plants that were not annotated in Dunnett's dataset to ensure the labelled data covered most of the parameter space of PV power plants in China. The total area of the PV power plants in China is about 897 km$^2$ from primary Dunnett's dataset and the area of the modified training regions was 1121 km$^2$. We then randomly sampled points within the training with a balanced quantity from humid and arid regions in China. We have put the statements in the revised manuscript in Section 2.1.3 for clearer future readership.

3. Writing should be taken more seriously as there are many writing and grammatical errors in the manuscript. For example, in line 201," we discovered that the EVI values of PV power plants in 2020 was strongly and positively linked with the that in 2013".

Response to comment 3: We have thoroughly gone through the manuscript to edit the writing and correct the grammatical errors.

4. The quality of the figures should be further improved. For example, the "North arrow" in Fig. 1 and Fig. 5 were missing. And in Fig. 2, what do 1-6 mean?

Response to comment 4: We have further improved all figures in the revised manuscript: we added the "North arrow" in Figure 1 and Figure 5. We are sorry for the unclear description of Figure 2. We refined the figure 2 with accurate description. We added the scale bar and coordinate in the Figure 2. The specific meanings of 1, 2, 3, 4, 5, and 6 are the 6 example sites to show the true-color images from Landsat-8, Sentinel-2, and Google Earth for visual interpretation.

---

## Author Response (AR2)

Author Response to the Reviewer Comments to the manuscript "Mapping photovoltaic power plants in China using Landsat, Random Forest, and Google Earth Engine" [essd-2022-16] submitted to Earth System and Science Data.

We thank the referee #3 for his/her constructive comments, which are listed below (shown in blue) along with our response (shown in black). Accordingly, we have incorporated all the requested changes in the revised manuscript. We have also added the data DOI (https://doi.org/10.5281/zenodo.6849477) to the end of Abstract.

**RC3: 'Comment on essd-2022-16', Anonymous Referee #3 and our response**

This a well conducted study, and the comments of R1 and R2 have improved the paper.
Response: We very much appreciate Reviewer#3's positive comments on our study.

I think one clarification is necessary before publication. R2 highlights the bias of relying on Dunnett's data, to which the authors respond: "So we further manually selected and edited the extent of different PV power plants that were not annotated in Dunnett's dataset" - does this mean there are some PV power plants in the training data that are not from this dataset? If so, it is not clear at the moment where these extra data come from. Are they manually searched for? Are they from some other national dataset?
Response: We are sorry for the unclear description of the training set. In addition to the Dunnett's dataset, we also manually collected data points for the training set. To do this, we firstly searched for the approximate distribution area of PV power plants through the web pages and then manually labelled the PV power plants on Google Earth high resolution images in 2017. These extra labelled PV power plants (approx. xx) are mainly located in eastern China, where PV power plants are rarely labelled in Dunnett's dataset.

We improved the sentence in Section 2.1.3 "*With high resolution Google Earth images of 2017, we further enriched the training dataset by manually selecting and labelling xx PV power plants over regions of eastern China, where PV power plants are rarely labelled in Dunnett's dataset.*

One of the objectives to the paper is to provide as replicable a process as possible for those without the computational means etc. so I feel like a full account of the training data is necessary.
Response: Thank you for the advice! We have uploaded our training points with training features for Landsat-8 imagery in this study as well as the labelled PV power plants to a data sharing website. It is publicly accessible at https://doi.org/10.5281/zenodo.6849477.

As an aside, I notice the RF model does poorly identifying "mountainous" PV sites, and I note that the authors purposefully excluded pixels with a >30deg slope. Could these two concepts be linked?
Response: Thanks for the comment. These two concepts are not really connected. The RF model didn't perform well for identifying "mountainous" PV sites mainly due to the unique installation spacing and installation angles of the solar panels in such area. Additionally, the mountainous terrain also affects the reflectance of the PV power plants, causing a distinct reflectance of PV power plants

in mountain areas from that in other flat areas. Therefore, the accuracy of our model's identification of PV power plants in mountainous areas is generally lower.

The reason we excluded pixels with the slope being over 30° for the analysis of this study was because the PV power plants are generally not built in locations with large slopes due to construction difficulty and soil erosion protection. As a result, we excluded very slopy areas that wouldn't have PV power plants to reduce the workload and computing costs of this study.